# Representational drift as a result of implicit regularization

Aviv Ratzon[1,2]*, Dori Derdikman[1], Omri Barak[1,2]

[1]Rappaport Faculty of Medicine, Technion - Israel Institute of Technology, Haifa, Israel; [2]Network Biology Research Laboratory, Technion - Israel Institute of Technology, Haifa, Israel

**Abstract** Recent studies show that, even in constant environments, the tuning of single neurons changes over time in a variety of brain regions. This representational drift has been suggested to be a consequence of continuous learning under noise, but its properties are still not fully understood. To investigate the underlying mechanism, we trained an artificial network on a simplified navigational task. The network quickly reached a state of high performance, and many units exhibited spatial tuning. We then continued training the network and noticed that the activity became sparser with time. Initial learning was orders of magnitude faster than ensuing sparsification. This sparsification is consistent with recent results in machine learning, in which networks slowly move within their solution space until they reach a flat area of the loss function. We analyzed four datasets from different labs, all demonstrating that CA1 neurons become sparser and more spatially informative with exposure to the same environment. We conclude that learning is divided into three overlapping phases: (i) Fast familiarity with the environment; (ii) slow implicit regularization; and (iii) a steady state of null drift. The variability in drift dynamics opens the possibility of inferring learning algorithms from observations of drift statistics.

## eLife assessment

This study presents a new and **important** theoretical account of spatial representational drift in the hippocampus. The evidence supporting the claims is **convincing**, with a clear and accessible explanation of the phenomenon. Overall, this study will likely attract researchers exploring learning and representation in both biological and artificial neural networks.

## Introduction

What do we mean when we say that the brain represents the external world? One interpretation is the existence of neurons whose activity is tuned to world variables. Such neurons have been observed in many contexts: place cells (*O'keefe and Nadel, 1979*; *O'Keefe and Dostrovsky, 1971*) – which are tuned to position in a specific context, visual cells (*Hubel and Wiesel, 1962*) – which are tuned to specific visual cues, neurons that are tuned to the execution of actions (*McNaughton et al., 1994*) and more. This tight link between the external world and neural activity might suggest that, in the absence of environmental or behavioral changes, neural activity is constant. In contrast, recent studies show that, even in constant environments, the tuning of single neurons to outside world variables gradually changes over time in a variety of brain regions, even long after good representations of the stimuli were achieved. This phenomenon has been termed *representational drift*, and has changed the way we think about the stability of memory and perception, but its driving forces and properties are still unknown (*Mankin et al., 2012*; *Ziv et al., 2013*; *Driscoll et al., 2017*; *Deitch et al., 2021*;

*For correspondence:
aviv.ratzon@hotmail.com

Competing interest: The authors declare that no competing interests exist.

**Figure 1.** Two types of possible movements within the solution space. (**A**) Two options of how drift may look in the solution space. Random walk within the space of equally good solutions that is either undirected (left) or directed (right). (**B**) The qualitative consequence of the two movement types. For an undirected random walk, all properties of the solution will remain roughly constant (left). For the directed movement there should be a given property that is gradually increasing or decreasing (right).

*Schoonover et al., 2021*; *Jacobson et al., 2018*) (see *Liberti et al., 2022*; *Sadeh and Clopath, 2022* for an alternative account).

There are at least two immediate theoretical questions arising from the observation of drift – why does it happen, and whether and how behavior is resistant to it (*Rule et al., 2019*; *Driscoll et al., 2022*)? One mechanistic explanation is that the underlying anatomical substrates are themselves undergoing constant change, such that drift is a direct manifestation of this structural morphing (*Ziv and Brenner, 2018*). A normative interpretation posits that drift is a solution to a computational demand, such as temporal encoding (*Rubin et al., 2015*), 'drop-out' regularization (*Aitken et al., 2022*), exploration of the solution space (*Kappel et al., 2015*), or re-encoding during continual learning (*Rule et al., 2019*). Several studies also address the resistance question, providing possible explanations on how behavior can be robust to such phenomena (*Rokni et al., 2007*; *Susman et al., 2019*; *Mongillo et al., 2017*; *Kossio et al., 2021*).

Here, we focus on the mechanistic question, and leverage analyses of drift statistics for this purpose. Specifically, recent studies suggest that representational drift in the CA1 is driven by active experience (*Khatib et al., 2023*; *Geva et al., 2023*). Namely, rate maps decorrelate more when mice are active for a longer time in a given context. This implies that drift is not just a passive process, but rather an active learning one. As drift seems to occur after an adequate representation has formed, it seems fitting to model it as a form of a continuous learning process.

This approach has been recently explored by *Qin et al., 2023*; *Pashakhanloo and Koulakov, 2023*. They considered continuous learning in noisy, overparameterized neural networks. Because the system is overparameterized, a manifold of zero-loss solutions exists. (*Qin et al., 2023*) showed that for feedforward neural networks (FNNs) trained using Hebbian learning with added parameter noise, units change their tuning over time. This was due to an *undirected* random walk within the manifold of solutions. The coordinated drift of neighboring place fields was used as evidence to support this view. The phenomenon of undirected motion within the space of solutions seems plausible, as all members of this space achieve equally good performance (*Figure 1A* left). However, there may be

other properties of the solutions (*Figure 1B*) that vary along this manifold, which could potentially bias drift in a certain direction (*Figure 1A* right). It is likely that the drift observed in experiments is a combination of both an undirected and directed movement. We will now introduce theoretical results from machine learning that support the possibility of directed drift.

Recent work provided a tractable analytical framework for the learning dynamics of Stochastic Gradient Descent (SGD) with added noise and an overparameterized regime (*Blanc et al., 2020*; *Li et al., 2021*; *Yang et al., 2023*). These studies showed that, after the network has converged to the zero-loss manifold, a second-order effect biases the random walk along a specific direction within this manifold. This direction reduces an implicit regularizer, determined by the type of noise the network is exposed to. The regularizer is related to the Hessian of the loss – a measure of the flatness of the loss landscape in the vicinity of the solutions. Since this directed movement is a second-order effect, its timescale is orders of magnitude larger than that of the initial convergence.

Consider a biological neural network performing a task. The machine learning (ML) implicit regularization mentioned above requires three components: an overparameterized regime, noise, and SGD. Both biological and artificial networks possess a large number of synapses, or parameters, and hence can reasonably be expected to be overparameterized. Noise can emerge from the external environment or from internal biological elements. It is not reasonable to assume that a precise form of gradient descent is implemented in the brain (*Bengio et al., 2015*), thereby casting doubt on the third element. Nevertheless, biologically plausible rules could be considered as noisy versions of gradient descent, as long as there is a coherent improvement in performance (*Liu et al., 2022*; *Marschall et al., 2020*). Motivated by this analogy, we explore representational drift in models and experimental data.

Because drift is commonly observed in spatially-selective cells, we base our analysis on a model which has been shown to contain such cells (*Recanatesi et al., 2021*). Specifically, we trained artificial neural networks on a predictive coding task in the presence of noise. In this task, an agent moves along a linear track while receiving visual input from the walls, such that the goal is to predict the subsequent input. We observed that hidden layer units became tuned to the latent variable, which is position, in accordance with previous results (*Recanatesi et al., 2021*). We continued training and found that in addition to the gradual change of tuning curves, similar to *Qin et al., 2023*, we witnessed that the fraction of active units decreased slowly while their tuning specificity increased. We show that these results align with experimental observations from the CA1 area - namely that when exposed to a novel environment, the number of active cells reduces while their tuning specificity increases long after the environment is already familiar. Finally, we demonstrated the connection between this sparsificiation effect and changes to the Hessian, in accordance with ML theory. That is, changes in activity statistics (sparseness) and representations (drift) are the signatures of the movement in solution space to a flatter area, until flatness saturates. We conclude that learning is divided into three overlapping phases: (i) Fast familiarity with the environment in which representations form; (ii) slow implicit regularization which can be recognized by changes in activity statistics; and (iii) a steady state of null drift in which representations gradually change.

## Results

### Spontaneous sparsification in a predictive coding network

To model representational drift in the CA1 area, we chose a simple model that could give rise to spatially-tuned cells (*Recanatesi et al., 2021*). In this model, an agent traverses a corridor while slightly modulating its angle with respect to the main axis (*Figure 2A*). The walls are textured by a fixed smooth noisy signal, and the agent receives this as input according to its current field of view. The model itself is a single hidden layer feedforward network, with the velocity and visual field as inputs. The desired output is the predicted visual input in the next time step. The model equations are given by:

$$\hat{y}_t = \sigma(x_t \mathbf{m}^T + \mathbf{b})\mathbf{n}^T, \tag{1}$$

$$\sigma(x) = max(0, x), \tag{2}$$

where $\mathbf{m}$ and $\mathbf{n}$ are the input and output matrices, respectively, $\mathbf{b}$ is the bias vector, and $\sigma$ is the ReLU activation function. The vector $\sigma(\mathbf{x}_t \mathbf{m}^T + \mathbf{b})$ constitutes the hidden layer, and each element in it is the

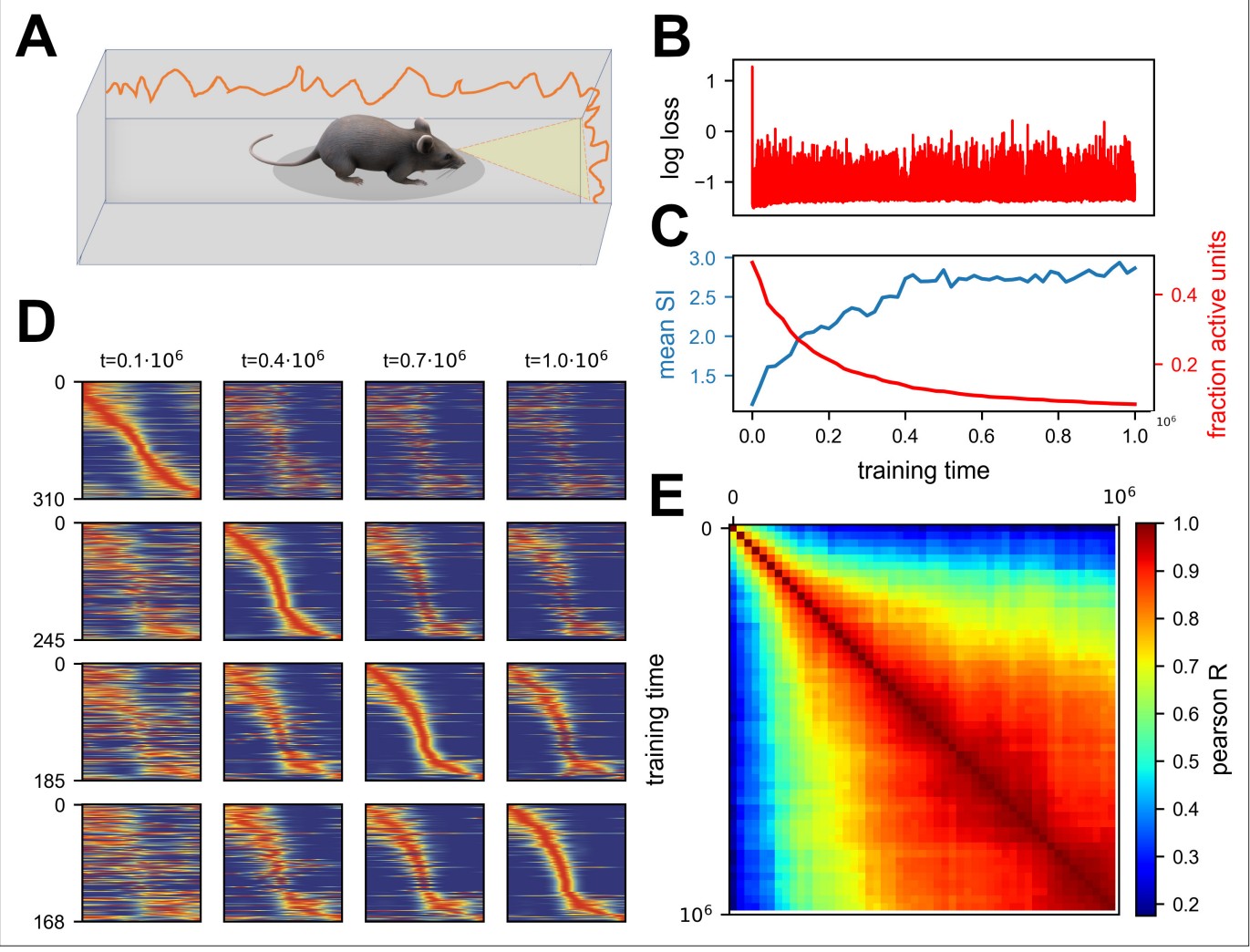

**Figure 2.** Continuous noisy learning leads to drift and spontaneous sparsification. (**A**) Illustration of an agent in a corridor receiving high-dimensional visual input from the walls. (**B**) Loss as a function of training steps (log scale). Zero loss corresponds to a mean estimator. Note the rapid drop in loss at the beginning, after which it remains roughly constant. (**C**) Mean spatial information (SI, blue) and fraction of units with non-zero activation for at least one input (red) as a function of training steps. (**D**) Rate maps sampled at four different time points (columns). Maps in each row are sorted according to a different time point. Sorting is done based on the peak tuning value to the latent variable. (**E**) Correlation of rate maps between different time points along training. Only active units are used.

activation of a given unit for the input at time $t$. The task is for the network's output, $\hat{\mathbf{y}}$, to match the visual input, $\mathbf{x}$ of the following time step, resulting in the following loss function:

$$f(\mathbf{m}, \mathbf{n}, \mathbf{b}) = \mathbb{E}_t(\hat{\mathbf{y}}_t - \mathbf{x}_{t+1})^2. \tag{3}$$

We train the network using Gradient Descent (GD), while adding update noise to the learning dynamics:

$$\theta_{\tau+1} = \theta_\tau - \eta \frac{\partial f(\theta_\tau)}{\partial \theta_\tau} + \xi_\tau^{update}, \tag{4}$$

where $\theta = (\mathbf{m}, \mathbf{n}, \mathbf{b})$ is the vectorized parameters-vector, $\tau$ is the current training step and $\xi_\tau^{update}$ is Gaussian noise. We let the network converge to a good solution, demonstrated by a loss plateau, and continue training for an additional period. Note that this additional period can be orders of magnitude longer than the initial training period. The network quickly converged to a low loss and stayed at the same loss during the additional training period (**Figure 2B**). Surprisingly, when looking at the activity of units within the hidden layer, we noticed that it slowly became sparse (see methods for definitions

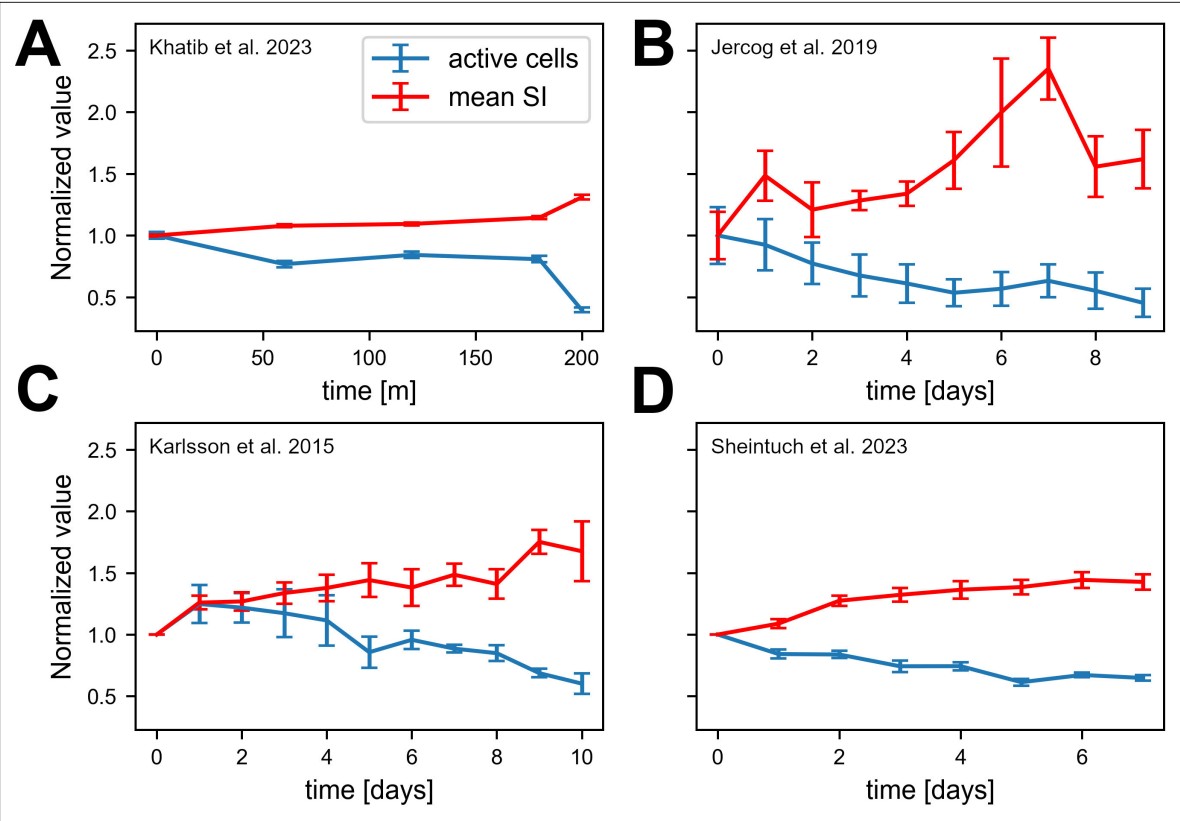

**Figure 3.** Experimental data consistent with simulations. Data from four different labs show sparsification of CA1 spatial code, along with an increase in the information of active cells. Values are normalized to the first recording session in each experiment. Error bars show standard error of the mean. (**A**) Fraction of place cells (slope=-0.0003 p < .001) and mean spatial information (SI) (slope=0.002, p < .001) per animal over 200 min (***Khatib et al., 2023***). (**B**) Number of cells per animal (slope=-0.052, p = .004) and mean SI (slope=0.094, p < .001) over all cells pooled together over 10 days. Note that we calculated the number of active cells rather than fraction of place cells because of the nature of the available data (***Jercog et al., 2019b***). (**C**) Fraction of place cells (slope=-0.048, p = .011) and mean SI per animal (slope=0.054, p < .001) over 11 days (***Karlsson and Frank, 2008***). (**D**) Fraction of place cells (slope=-0.026, p < .001) and mean SI (slope=0.068, p < .001) per animal over 8 days (***Sheintuch et al., 2023***).

of sparseness). This sparsification did not hurt performance, because individual units became more informative (***Figure 2C***), as quantified by the average Spatial Information (SI, see methods). When looking at the rate maps of units, i.e., their tuning to position, one can observe an image similar to representational drift observed in experiments (***Ziv et al., 2013***) – namely that neurons changed their tuning over time (***Figure 2D***). Additionally, their tuning specificity increased in accordance with the SI increase. By observing the correlation matrix of the rate maps over time, it is apparent that there was a gradual change that slowed down (***Figure 2E***). To summarize, we observed a spontaneous sparsi-fication over a timescale much longer than the initial convergence, without introducing any explicit regularization.

At first glance, these results might seem inconsistent with the experimental descriptions of drift reported in the literature in which all metrics are stationary while only representations change (***Ziv et al., 2013***). We suggest that there exists an intermediate phase between initial familiarity and stationary activity metrics, which is consistent with the notion of drift, that exhibits a gradual change in activity statistics. It seems that most experimental paradigms require a long pre-exposure, longer than needed to become fully familiarized with the environment, thus missing the suggested effect. We thus analyzed four datasets, from four different labs, in which we believe that the familiarization stage was shorter than in other studies. Some of the analyses were present in the original papers, and others are novel using publicly available data. Three experiments start from a novel environment and one from a relatively short pre-exposure. As shown in ***Figure 3***, all datasets are consistent with our simulations - namely that the fraction of active cells reduces while the mean SI per cell increases over

a long timescale. See Methods section for a full description of the data sets and analyses, along with another paper (*Geva et al., 2023*) in which activity statistics are stationary, for comparison.

## Generality of the phenomenon

The theoretical considerations (*Blanc et al., 2020*; *Li et al., 2021*), simulation results, and experimental results from multiple labs all suggest very general and robust phenomena.

To explore the sensitivity of our results to specific modeling choices, we systematically varied many of them. Specifically, we replaced tasks (see below) and simulated different activation functions. Perhaps most important, we varied the learning rules, as SGD is not a biologically plausible one. We used both Adam (*Kingma and Ba, 2014*) and RMSprop (*Hinton et al., 2012*), from the ML literature. We also used Stochastic Error-Descent (SED) (*Cauwenberghs, 1992*), which does not require gradient calculation and is more biologically plausible (6). For SGD, we also ran simulations with label noise instead of update noise. Finally, we replaced the task with either a simplified predictive coding, random mappings, or smoothed random mapping. The motivation for different tasks is twofold. First, there is some arbitrariness in the predictive task we chose. Second, the interpretation of drift as a result of continuous learning in the presence of noise, suggests that this effect goes beyond the specific phenomenon of place cells. That is, drift within the solution space should occur in every type of task and scenario, and could be identified outside the scope of spatial representations. All 1151 simulations, except a negligible few, demonstrated an initial, fast phase of convergence to low loss, followed by a much slower phase of directed random motion within the low-loss space.

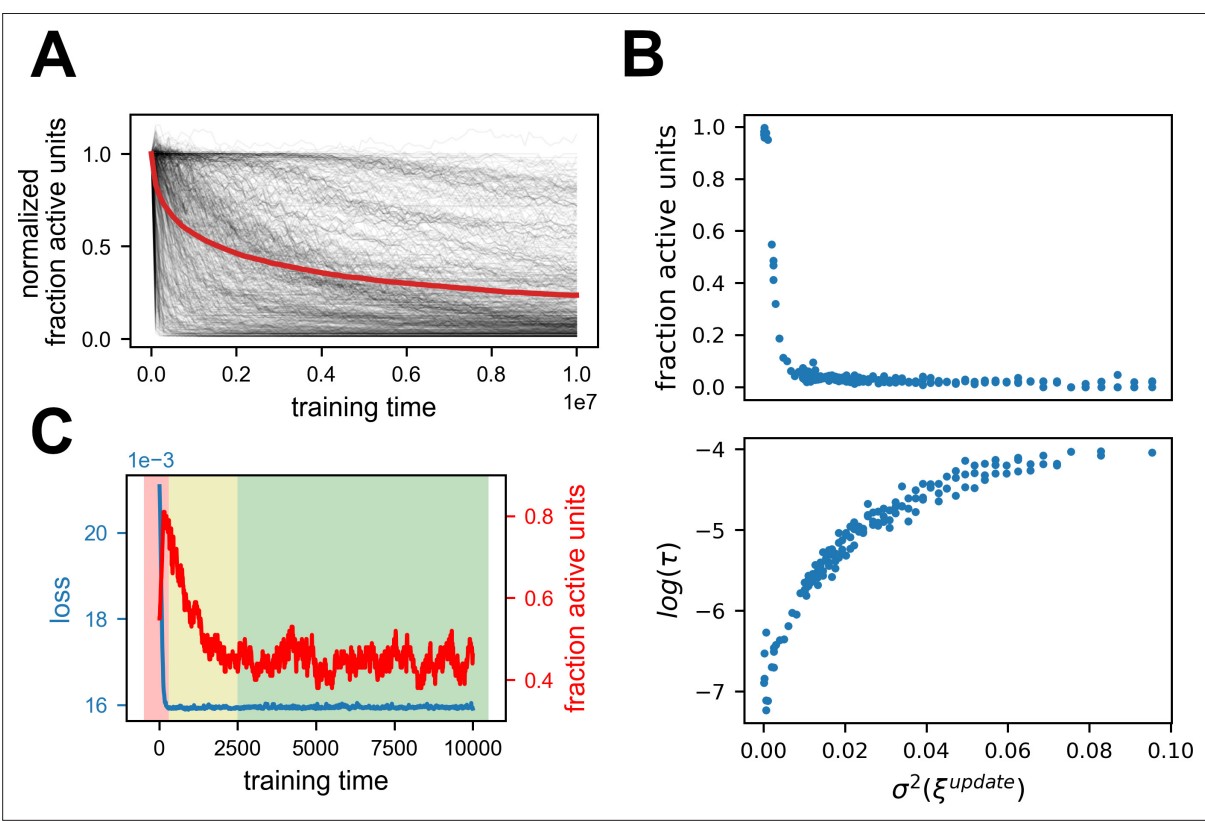

**Figure 4.** Generality of the results. Summary of 616 simulations with various parameters, excluding stochastic gradient descent (SGD) with label noise (see *Table 2*). (**A**) Fraction of active units normalized by the first timestep for all simulations. Red line is the mean. Note that all simulations exhibit a stochastic decrease in the fraction of active units. See *Figure 4—figure supplement 1* for further breakdown. (**B**) Dependence of sparseness (top) and sparsification time scale (bottom) on noise amplitude. Each point is one of 178 simulations with the same parameters except noise variance. (**C**) Learning a similarity matching task with Hebbian and anti-Hebbian learning using published code from *Qin et al., 2023*. Performance of the network (blue) and fraction of active units (red) as a function of training steps. Note that the loss axis does not start at zero, and the dynamic range is small. The background colors indicate which phase is dominant throughout learning (1 - red, 2 - yellow, 3 - green).

The online version of this article includes the following figure supplement(s) for figure 4:

**Figure supplement 1.** Noisy learning leads to spontaneous sparsification.

The results of the simulations supported our main conclusion – sparsification dynamics were not sensitive to most of the parameters. For almost all of the simulations, excluding SGD with label noise, the fraction of active units gradually reduced, long after the loss converged (*Figure 4A*, *Figure 4—figure supplement 1* for further breakdown). For label noise, a slow directed effect was observed but the dynamics were qualitatively different – as predicted by theory (*Li et al., 2021*) and explained in the next section. The fraction of active units did not reduce as much, but the activity of the units did sparsify (*Figure 5—figure supplement 1*). One qualitative difference observed between simulations was that the timescales could vary by orders of magnitude as a function of the noise scale (*Figure 4B* bottom, see methods for details). Additionally, apart from simulations that did not converge due to too large timescales, the final sparsity was the same for all networks with the same parameters (*Figure 4B* top), in accordance with results from *Qin et al., 2023*. In a sense, once noise is introduced, the network is driven to maximal sparsification in a stochastic manner. For Adam, RMSprop and SED sparsification ensued in the absence of any added noise. For SED the explanation is straightforward, as the parameter updates are driven by noise. For Adam and RMSprop, we suggest that in the vicinity of the zero-loss manifold, the second moment acts as noise. In some cases, the networks quickly collapsed to a sparse solution, most likely as a result of the learning rate being too high, in relation to the input statistics (*Mulayoff et al., 2021*). Importantly, for GD without noise, there was no change after the initial convergence.

As a further test of the generality of this phenomenon, we consider the recent simulation from *Qin et al., 2023* in which representational drift was shown. The learning rule used in this work was very different from the ones we applied, and more biologically plausible. We simulated that network using the published code and found the same type of dynamics as shown above. Namely, the network initially converged to a good solution followed by a longer period of sparsification (*Figure 4C*). Note that in the original publication (*Qin et al., 2023*) the focus was on the stage following this sparsification, in which the network indeed maintained a constant fraction of active cells.

In conclusion, we see that noisy learning leads to three phases under rather general conditions. The first phase is the learning of the task and convergence to the manifold of low-loss solutions. The second phase is directed movement on this manifold, driven by a second-order effect of implicit regularization. The third phase is an undirected random walk within the sub-manifold of low loss and maximum regularization. *Table 1* summarizes these three phases and their signature in network metrics. It is important to note that these are not consecutive phases but rather overlapping ones – they occur simultaneously, but due to their different time scales one can identify when each phase is dominant. A rough delineation of when each phase is dominant can be seen in *Figure 4C* background colors – in the first phase (red) the loss function converged, in the second phase (yellow) the fraction of active units reduced substantially. In the third phase (red) both were stationary but the tuning of units continued to change, as shown in the original paper (*Qin et al., 2023*). We speculate that most experimental studies about drift demonstrated only the third phase of null drift, because they familiarized the animals to the environment for a substantial time period prior to recording. We refer to the second phase as a type of drift, because it happens after learning has finished and also features a gradual change in representations.

## Mechanism of sparsification

What are the mechanisms that give rise to this observed sparsification? As illustrated in *Figure 1*, solutions in the zero-loss manifold have identical loss, but might vary in some of their properties. The authors of *Blanc et al., 2020* suggest that noisy learning will slowly increase the flatness of the loss landscape in the vicinity of the solution. This can be demonstrated with a simple example. Consider a two-dimensional loss function. The function is shaped like a valley with a continuous one-dimensional zero-loss manifold at its bottom (*Figure 5A*). Crucially, the loss on this one-dimensional manifold is

**Table 1.** The three phases of noisy learning.

| Phase | Duration | Performance | Activity statistics | Representations |
|---|---|---|---|---|
| learning of task | short | changing | changing | changing |
| directed drift | long | stationary | changing | changing |
| null drift | endless | stationary | stationary | changing |

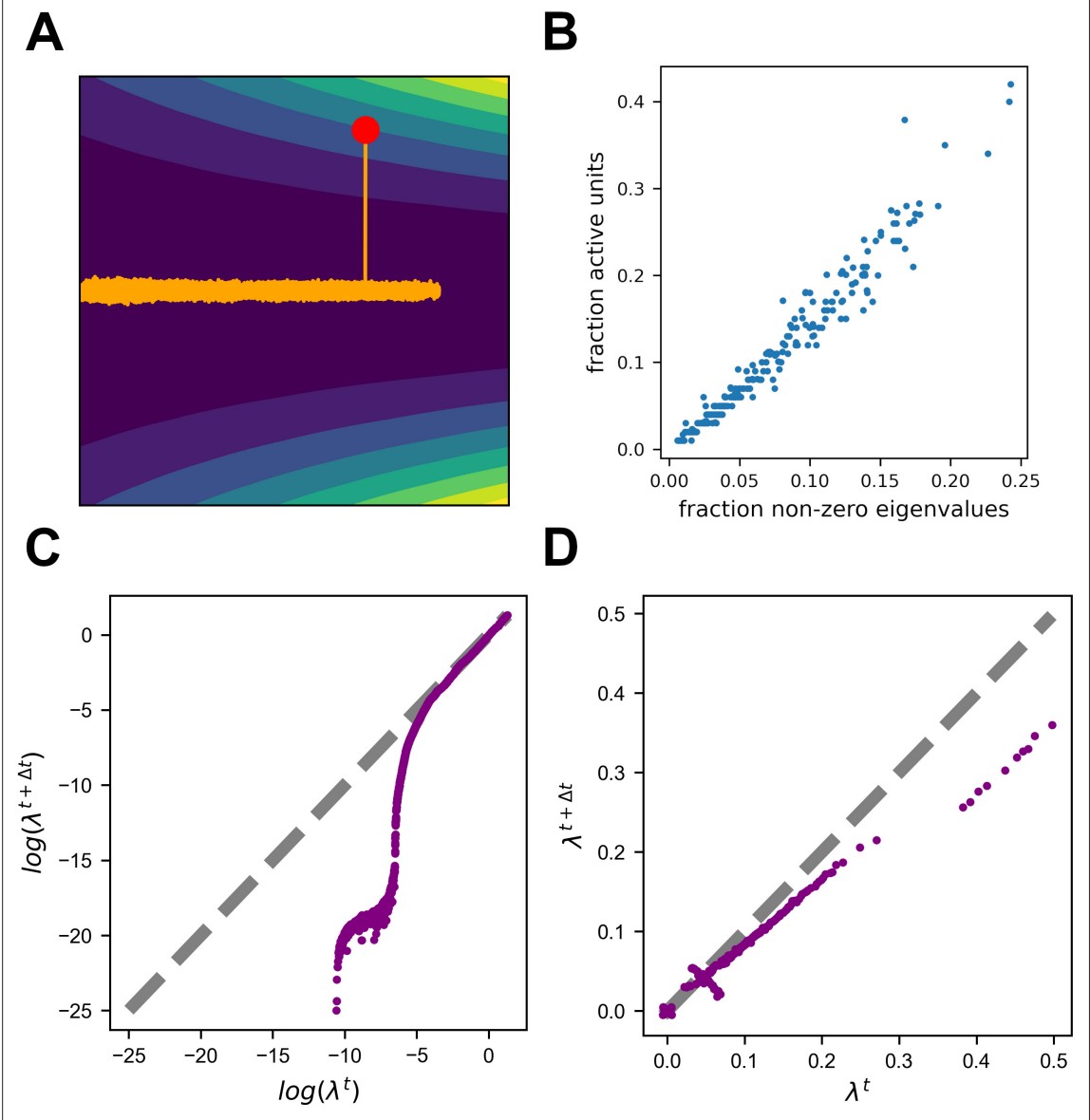

**Figure 5.** Noisy learning leads to a flat landscape. (**A**) Gradient Descent dynamics over a two-dimensional loss function with a one-dimensional zero-loss manifold (colors from blue to yellow denote loss). Note that the loss is identically zero along the horizontal axis, but the left area is flatter. The orange trajectory begins at the red dot. Note the asymmetric extension into the left area. (**B**) Fraction of active units is highly correlated with the number of non-zero eigenvalues of the Hessian. (**C**) Update noise reduces small eigenvalues. Log of non-zero eigenvalues at two consecutive time points for learning with update noise. Note that eigenvalues do not correspond to one another when calculated at two different time points, and this plot demonstrates the change in their distribution rather than changes in eigenvalues corresponding to specific directions. The distribution of larger eigenvalues hardly changes, while the distribution of smaller eigenvalues is pushed to smaller values. (**D**) Label noise reduces the sum over eigenvalues. Same as (**C**), but for actual values instead of log.

The online version of this article includes the following figure supplement(s) for figure 5:

**Figure supplement 1.** Label and update noise impose different regularization over the Hessian with distinct signatures in activity statistics.

exactly zero, while the vicinity of the manifold becomes systematically flatter in one direction. We simulated gradient descent with added noise on this function from a random starting point (red dot). The trajectory quickly converged to the zero-loss manifold, and began a random walk on it. This walk was clearly biased towards the flatter area of the manifold, as can be seen by the spread of the trajectory. This bias could be comprehended by noting that the gradient was orthogonal to the contour lines of the loss, and therefore had a component directed towards the flat region.

In higher dimensions, flatness is captured by the eigenvalues of the Hessian of the loss. Because these eigenvalues are a collection of numbers, different scenarios could lead to minimizing different aspects of this collection. Specifically, according to *Blanc et al., 2020*, update noise should regularize the sum of the log of the non-zero eigenvalues while label noise should do the same for the sum of eigenvalues. In our predictive coding example, where update noise was added, each inactivated unit translates into a set of zero-rows in the Hessian (see methods), and thus also into a set of zero-eigenvalues (*Figure 5B*). The slope of the regularizer approaches infinity as the eigenvalue approaches zero, and thus small eigenvalues are driven to zero much faster than large eigenvalues (*Figure 5C*). So in this case, update noise leads to an increase in the number of zero eigenvalues, which are manifested as a sparse solution. Another, perhaps more intuitive, way to understand these results is that units below the activation threshold are insensitive to noise perturbations. In other scenarios, in which we simulated with label noise, we indeed observed a gradual decrease in the sum of eigenvalues (*Figure 5D*). For a more intuitive demonstration of this phenomenon, see *Figure 5— figure supplement 1*.

## Discussion

We showed that representational drift could arise from ongoing learning in the presence of noise, after a network has already reached good performance. We suggest that learning is divided into three overlapping phases: a fast initial phase, where good performance is achieved, a second slower phase in which *directed* drift along the low-loss manifold leads to an implicit regularization and finally, a third *undirected* phase ensues once the regularizer is minimized. In our results, the directed component was associated with sparsification of the neural code. We verified the existence of this phenomenon in experimental data from four different labs. It is important to note that sparseness was related to flatness of the loss landscape in the specific case of a single hidden layer feedforward neural network and update or label noise. For other architectures and noise types, the change in activity statistics will most likely be different and calls for further work. The CA1 region is known to have little recurrent connections, which possibly explains why these results match.

Interpreting drift as a learning process has recently been suggested by *Qin et al., 2023*; *Pashakhanloo and Koulakov, 2023*. Both studies focused on the final phase in which the statistics of the representations were constant. Experimentally, (*Deitch et al., 2021*) reported a decrease in activity at the beginning of the experiment, which they suggested was correlated with some behavioral change, but we believe it could also be a result of the directed drift phase. (*Nguyen et al., 2022*) also reported a slow directed change in representation long after familiarity with the stimuli. There is another consequence of the timescale separation. Unlike in the setting of drift experiments, natural environments are never truly constant. Thus, it is possible that the second phase of learning never stops because the task is slowly changing. This would imply that the second, directed, phase may be the natural regime in which neural networks reside.

Here, we reported directed drift in the space of solutions of neural networks. This drift could be observed by examining changes to the representation of external world variables, and hence is related to the phenomenon of representational drift. Note, however, that representations are not a full description of a network's behavior (*Brette, 2019*). The statistics of representational changes can be used as a window into changes of network dynamics and function.

The phenomenon of directed drift is very robust to various modeling choices, and also consistent with recent theoretical results (*Blanc et al., 2020*; *Li et al., 2021*) The details of the direction of the drift, however, are dependent on specific choices. Specifically, which aspects of the Hessian are minimized during the second phase of learning, as well as the timescale of this phase, depend on the specifics of the learning rule and the noise in the system. This suggests an exciting opportunity – inferring the learning rule of a network from the statistics of representational drift.

Our explanation of drift invoked the concept of a low-loss manifold – a family of network configurations that have nearly identical performance on a task. The definition of low-loss, however, depends on the specific task and context analyzed. Challenging a system with new inputs could dissociate two configurations that otherwise appear identical (*Turner et al., 2021*). It will be interesting to explore whether various environmental perturbations could uncover the motion along the low-loss manifold in the CA1 population. An important subject relating to perturbations is that of remapping – the phenomenon in which place cells change their tuning in response to a change in the environment or

change in context. One can, therefore, speculate that, as the network moves to flatter areas of the loss landscape, becoming more robust to noise and thus to perturbations, the probability for remapping given the same environmental change will systematically decrease. A functional interpretation of remapping as latent state (context) inference is given by *Sanders et al., 2020*. The authors also summarize results from a series of morph experiments and offer a prediction similar to ours (Fig. 7 there) – that discrepancies between remapping probabilities can be explained as testing at different points of training. Beyond such abstract models, for future work, it is also possible to mechanistically model multiple environments and study remapping probabilities through them (*Low et al., 2023*).

Machine learning has been suggested as a model tool for neuroscience research (*Richards et al., 2019*; *Marblestone et al., 2016*; *Saxe et al., 2021*). However, the implicit regularization in ML has not been studied to explain representational drift in neuroscience, and may have been done without awareness of this phenomenon. It's worth noting that this isn't a phenomenon specific to neural networks, but rather a general property of overparameterized systems that optimize a cost function. Importing insights from this domain into neuroscience shows the utility of studying general phenomena in systems that learn. For example, another complex learning system in which a similar idea has been proposed is evolution – 'survival of the flattest' suggests that, under a high mutation rate, the fittest replicators are not just the ones with the highest fitness, but also with a flat fitness function which is more robust to mutations (*Codoñer et al., 2006*). One can hope that more such insights will arise as we open our eyes.

### Code availability

The code for this project is available at https://github.com/Aviv-Ratzon/DriftReg, (copy archived at *Aviv-Ratzon, 2024*).

## Materials and methods
### Predictive coding task

The agent is moving in an arena of size $(L_x, L_y)$, with constant velocity in the $y$ direction of $V_0$. The agent's heading direction is θ and it changes at every time step by $\Delta\theta \sim G(0, \sigma_\theta^2)$, the agent's visual field has an angle $\theta_{vis}$ and is represented as a vector of size $L_{vis}$. The texture of the walls is generated from a random Gaussian vector of size $L_{walls} = 2(L_x + L_y)L_{vis}$, smoothed with a Gaussian filter with $\sigma^2 = K_{smooth}L_{walls}$. At each time step the agent receives the visual input from the walls, determined by the intersection points of its visual field with the walls. When the agent reaches a distance of $L_yL_{buffer}$ from the wall, it turns in the opposite direction.

### Tuning properties of units

For each unit we calculated a tuning curve. We divided the arena into 100 equal bins and computed the number of time steps in each bin and the mean unit activation. We then obtained the tuning curve

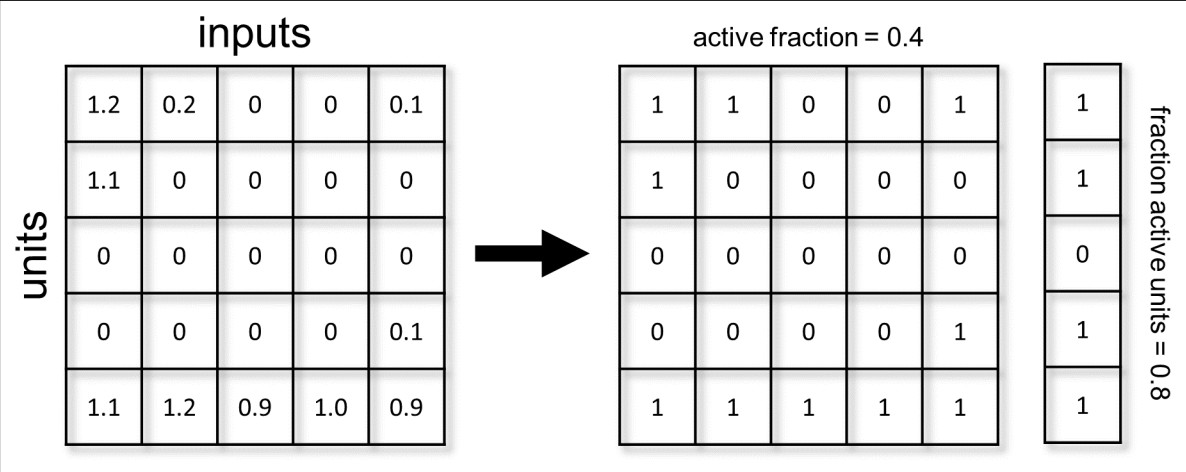

**Figure 6.** Illustration of sparsity metrics.

**Table 2.** Parameter ranges for random simulations.

| Parameter | Possible values |
|---|---|
| learning algorithm | {SGD, Adam, SED} |
| noise type | {update, label} |
| number of samples | *O'keefe and Nadel, 1979*; *Susman et al., 2019* |
| initialization regime | {lazy, rich} |
| task | {abstract predictive, random, random smoothed} |
| input dimension | *O'keefe and Nadel, 1979*; *Susman et al., 2019* |
| output dimension | *O'keefe and Nadel, 1979*; *Susman et al., 2019* |
| noise variance (label/update) | [0.1,1]/[0.01,0.1] |
| hidden layer size | 100 |

by dividing the mean activity for each bin by the occupancy. We treated movement in each direction as a separate location. We calculated the spatial information (SI) of the tuning curves for each unit:

$$SI = \sum_i p_i \frac{r_i}{\bar{r}} \log_2 \frac{r_i}{\bar{r}} \tag{5}$$

where $i$ is the index of the bin, $p_i$ is the probability of being in the bin, $r_i$ is the value of the tuning curve in the bin and $\bar{r}$ is the unit's mean activity rate. Active unit was defined as a unit with non-zero activation for at least one input.

## Measuring sparsity

We measure sparsity using two metrics: Active fraction, and fraction active units. These can be calculated from the activation matrix (*Figure 6*), where each row corresponds to a single unit, each column corresponds to an input and the values of the cells are the activations of a given unit for a given input. We can then binarize this matrix, giving a value of 1 to cells with non-zero activations. The active fraction is the mean over all cells of the matrix. The fraction of active units is the fraction of rows with at least one non-zero value. Note that 'units' refers to hidden layer units.

## Simulations

For the random simulations, we train each network for $10^7$ training steps while choosing random learning algorithm and parameters. The ranges and relevant values of parameters are specified in *Table 2*. For Adam and SED there was no added noise.

## Sparsification timescale

To produce *Figure 4B* bottom, we fitted an exponential curve over the curve of fraction of active units for each simulation and extracted the time constant of the exponential. To simplify this fit, we shifted each curve such that it plateaus at zero. We also clipped the length of the curves at the point in which they reach 90% of their final value to avoid fitting noise in the plateau after convergence. Note that we did this calculation to a subset of 178 simulations with the same parameters and varying noise scale. We chose this set of parameters because it exhibited a relatively 'nice' exponential curve. As can be seen in *Figure 4A*, some simulations exhibit a long plateau in the fraction of active units followed by a sudden start of the sparsification process. This type of plateau followed by a phase transition was described in previous works (*Saxe et al., 2013*; *Schuessler et al., 2020*). For some of the simulations finding the timescale of sparsification is not straightforward and rather noisy, thus for simplicity sake, we chose to calculate it only over the mentioned subset.

## Stochastic error descent

The equation for parameter updates under this learning rule is given by:

$$\theta_{\tau+1} = \theta_\tau - \eta(f(\theta_\tau + \xi_\tau) - f(\theta_\tau))\xi_\tau \tag{6}$$

**Table 3.** Description of experimental datasets.

| | Khatib et al., 2023 | Jercog et al., 2019b | Karlsson et al., 2015 | Sheintuch et al., 2023 | Geva et al., 2023 |
|---|---|---|---|---|---|
| Familiarity | 3–5 days | novel | novel | novel | 6–9 days |
| Species | mice | mice | rats | mice | mice |
| # Animals | 8 | 12 | 9 | 8 | 8 |
| Recordings days | 1 day | 10 days | max. 11 days | 10 days | 10 days |
| Session length | 200 min | 40 min | 15–30 min | 20 min | 20 min |
| Recording type | calcium imaging | electrophysiology | electrophysiology | calcium imaging | calcium imaging |
| Arena | linear track | square or circle | W-shaped | linear or L-shaped track | linear track |
| Activity metric | fraction of place cells decrease | number of active cells decrease | fraction of place cells decrease | fraction of place cells decrease | fraction of place cell stationary |
| Mean SI change | increase | increase | increase | increase | stationary |

In this learning rule, the parameters are randomly perturbed at each training step by a Gaussian noise denoted by $\xi_\tau$ and then updated in proportion to the change in loss.

## Experimental data

We present here a detailed description of the analyses performed for each dataset. *Table 3* summarizes the differences between them, along with an additional publication about CA1 drift *Geva et al., 2023* in which stationary statistics were reported. The p-values for the regression slopes were calculated using a t-test where the null hypothesis is that the coefficient is equal to zero. The regression included an intercept parameter.

*Khatib et al., 2023* - The results presented here were also shown in the paper, and a full description is available there. Only frames where the mice moved faster than $1\frac{cm}{s}$ were analyzed. We calculated the fraction of place cells out of all recognized cells, place cells were classified using a shuffle test. We also used the published code from *Sheintuch et al., 2022* to verify that the increase in SI is sustained under bias corrections. Note that we treated the linear track as one-dimensional and separated the two different running directions, bins were $4cm$ in length. The metrics were averaged over cells pooled together from all animals. Data is not publicly available.

*Jercog et al., 2019a* - The results presented here are novel. The published data features rate maps and full trajectories, without spike data. Only neurons with an average activity rate of over $0.1Hz$ were included. Because of this, we calculated only the number of active neurons each day rather than the fraction and could not classify which were place cells. We calculated the binned occupancy maps from the trajectories and used them with the published rate maps to calculate SI. Data is available at: https://crcns.org/data-sets/hc/hc-22/about-hc-22.

*Karlsson et al., 2015* - The results presented here are novel. The data features spikes and trajectories. We filtered the data to include time bins when the animals moved faster than $1\frac{cm}{s}$ and only neurons from CA1. We used the published code from *Sheintuch et al., 2022* to calculate the fraction of place cells along with SI, and verified that the increase in SI is sustained under bias corrections. For the occupancy map we treated the entire W-shaped arena as a square and used bins of approximately $4cm$ length. This was done for the sake of simplifying the analysis, a more accurate method would be to consider separately each linear track and movement direction. Note that in this dataset animals spent varying amounts of time in two different environments. We pooled together for each animal the data of cells from either environment according to experience time in them. For example, if the animal visited environment A on day 4 for the 4th time and had 20 active cells, and visited environment B on day 6 for the 4th time and had 30 active cells, we pooled together the entire 50 cells for day 4 of this animal. Data is available at: https://crcns.org/data-sets/hc/hc-6/about-hc-5.

*Sheintuch et al., 2023* - The results presented here were also shown in the paper. The data features various metrics calculated from the activity. We averaged the fraction of place cells and mean SI over animals for each day. Data is available at: https://github.com/zivlab/cell_assemblies, (copy archived at *zivlab, 2023*).

## Label noise

Label noise is introduced to the loss function given by the following formula:

$$f(\theta_\tau) = \mathbb{E}_t(\hat{\mathbf{y}}_t - \mathbf{x}_{t+1} + \xi_\tau^{label})^2, \tag{7}$$

where $\xi_\tau^{label}$ is Gaussian noise.

## Gradient descent dynamics around the zero-loss manifold

The function we used for the two-dimensional example was given by:

$$L(x, y) = (xy)^2, \tag{8}$$

which has zero loss on the $x$ and $y$ axes. For small enough update noise, GD will converge to the vicinity of this manifold (the axes). We consider a point on the $x$-axis: $(x_0, 0)$, and calculate the direction of the gradient near that point. Because we are interested in motion along the zero-loss manifold, we consider a small perturbation in the orthogonal direction $(x_0, 0 + \Delta y)$ where $x_0 \gg 1$ and $|\Delta y| \ll 1$. Any component of the gradient in the $x$ direction will lead to motion along the manifold. The update step at this point is given by:

$$-\nabla L(x_0, 0 + \Delta y) = -2x_0 \begin{pmatrix} (\Delta y)^2 \\ x_0 \Delta y \end{pmatrix} \tag{9}$$

One can observe that the step has a large component in the $y$ direction, quickly returning to the manifold. There is also a smaller component in the $x$ direction, reducing the value of $x$. Reducing $x$ also reduces the Hessian's eigenvalues:

$$H_L(x_0, 0) = 2 \begin{pmatrix} 0 & 0 \\ 0 & x_0^2 \end{pmatrix} \tag{10}$$

$$\lambda_{1,2} = \{0, x_0^2\}, v_{1,2} = \left\{ \begin{pmatrix} 1 \\ 0 \end{pmatrix}, \begin{pmatrix} 0 \\ 1 \end{pmatrix} \right\} \tag{11}$$

Thus, it becomes clear that the trajectory will have a bias that reduces the curvature in the $y$ direction.

For general loss functions and various noise models, rigorous proofs can be found in *Blanc et al., 2020*, and a different approach can be found in *Li et al., 2021*. Here, we will briefly outline the intuition for the general case. Consider again the update rule for GD:

$$\theta \leftarrow \theta - \eta \nabla L(\theta). \tag{12}$$

In order to understand the dynamics close to the zero-loss manifold, we consider a point θ, for which $L(\theta) = 0$ expand the loss around it:

$$L(\theta + \delta\theta) = L(\theta) + \nabla^T L(\theta)\delta\theta + \frac{1}{2}\delta\theta^T H \delta\theta. \tag{13}$$

We can then take the gradient of this expansion with respect to θ:

$$\nabla_\theta L(\theta + \delta\theta) = \nabla_\theta L(\theta) + \nabla_\theta \nabla_\theta^T L(\theta)\delta\theta + \nabla_\theta(\frac{1}{2}\delta\theta^T H \delta\theta) \tag{14}$$

$$= 0 + H\delta\theta + \nabla_\theta(\frac{1}{2}\delta\theta^T H \delta\theta). \tag{15}$$

The first term is zero, because the gradient is zero on the manifold. The second term is the largest one, as it's linear in $\delta\theta$. Note that the Hessian matrix has zero eigenvalues in directions on the zero-loss manifold, and non-zero eigenvalues in other directions. Thus, the second term corresponds to projecting $\delta\theta$ in a direction that is orthogonal to the zero-loss manifold. The third term can be interpreted as the gradient of some auxiliary loss function. Thus, we expect gradient descent to minimize this new loss, which corresponds to a quadratic form with the Hessian. This is the reason for the implicit regularization along the manifold. Note that the auxiliary loss function is defined by $\delta\theta$, and thus different noise statistics will correspond, on average, to different implicit regularizations. In conclusion, the update step will have a large component that moves the parameter vector towards the zero-loss manifold, and a small component that moves the parameter vector on the manifold in a direction that minimizes some measure of the Hessian.

### Hessian and sparseness

In the main text, we show that the implicit regularization of the Hessian leads to sparse representations. Here, we show this relationship for a single-hidden layer feed-forward neural network with ReLU activation and Mean Squared Error loss:

$$f(\mathbf{x}_i) = \sigma(\mathbf{x}_i\mathbf{m}^T + b)\mathbf{n}^T \tag{16}$$

The gradient and Hessian at the zero-loss manifold are given by *Nacson et al., 2023*:

$$\nabla_\theta f(\mathbf{x}_i) = \begin{pmatrix} \frac{\partial f}{\partial \mathbf{m}} \\ \frac{\partial f}{\partial \mathbf{b}} \\ \frac{\partial f}{\partial \mathbf{n}} \end{pmatrix} = \begin{pmatrix} n \odot \mathbb{1}(x_i;\theta) \otimes x_i \\ n \odot \mathbb{1}(x_i;\theta) \\ (x_i \cdot n^T + b) \odot \mathbb{1}(x_i;\theta) \end{pmatrix} \tag{17}$$

$$\nabla_\theta^2 L(\mathbf{x};\theta) = \sum_i \nabla_\theta f(\mathbf{x}_i)\nabla_\theta f(\mathbf{x}_i)^T, \tag{18}$$

where $\mathbb{1}(\mathbf{x}_i;\theta)$ is an indicator vector denoting whether each unit is active for some input $\mathbf{x}_i$. Sparseness means that a unit has become inactive for all inputs. All the partial derivatives of input, output and bias weights associated with such a unit are zero, and thus the relevant rows of the Hessian are zero as well. Thus, every inactive unit leads to several zero eigenvalues.

## Acknowledgements

We thank Ron Teichner and Kabir Dabholkar for their comments on the manuscript. This research was supported by the ISRAEL SCIENCE FOUNDATION (grants Nos. 2655/18 and 2183/21 to DD, and 1442/21to OB), by the German-Israeli Foundation (GIF I-1477–421.13/2018) to DD, by a grant from the US-Israel Binational Science Foundation (NIMH-BSF CRCNS BSF:2019807, NIMH:R01 MH125544-01 to DD), by an HFSP research grant (RGP0017/2021) to OB, A Rappaport Institute Collaborative research grant to DD, by Israel PBC-VATAT and by the Technion Center for Machine Learning and Intelligent Systems (MLIS) to DD and OB.

## Additional information

### Funding

| Funder | Grant reference number | Author |
| --- | --- | --- |
| Israel Science Foundation | grants Nos. 2655/19 | Dori Derdikman |
| Israel Science Foundation | grants Nos. 1442/21 | Omri Barak |
| German-Israeli Foundation for Scientific Research and Development | GIF I-1477-421.13/2018 | Dori Derdikman |

| Funder | Grant reference number | Author |
|---|---|---|
| US-Israel Binational Science Foundation | NIMH-BSF CRCNS BSF:2019807 | Dori Derdikman |
| Human Frontier Science Program | RGP0017/2021 | Omri Barak |
| Rappaport Institute Collaborative research grant | | Dori Derdikman |
| Israel PBC-VATAT and by the Technion Center for Machine Learning and Intelligent Systems | | Dori Derdikman Omri Barak |
| Israel Science Foundation | 2183/21 | Dori Derdikman |
| US-Israel Binational Science Foundation | NIMH:R01 MH125544- 372 01 | Dori Derdikman |

The funders had no role in study design, data collection and interpretation, or the decision to submit the work for publication.

## Author contributions

Aviv Ratzon, Conceptualization, Data curation, Formal analysis, Validation, Investigation, Visualization, Methodology, Writing - original draft, Project administration, Writing - review and editing; Dori Derdikman, Conceptualization, Resources, Data curation, Supervision, Funding acquisition, Methodology, Writing - review and editing; Omri Barak, Conceptualization, Resources, Supervision, Funding acquisition, Investigation, Methodology, Writing - review and editing

### Author ORCIDs
Aviv Ratzon ![ORCID] http://orcid.org/0009-0000-7648-9744
Dori Derdikman ![ORCID] http://orcid.org/0000-0003-3677-6321
Omri Barak ![ORCID] http://orcid.org/0000-0002-7894-6344

Reviewer #1 (Public Review): https://doi.org/10.7554/eLife.90069.3.sa1
Reviewer #2 (Public Review): https://doi.org/10.7554/eLife.90069.3.sa2
Reviewer #3 (Public Review): https://doi.org/10.7554/eLife.90069.3.sa3
Author response https://doi.org/10.7554/eLife.90069.3.sa4

# Additional files

## Supplementary files
• MDAR checklist

## Data availability
All the code and figure data to reproduce our manuscript is available at our GitHub repository (https://github.com/Aviv-Ratzon/DriftReg, copy archived at *Aviv-Ratzon, 2024*). Three of the four experimental datasets are publicly available and referred to in the methods section.

The following previously published datasets were used:

| Author(s) | Year | Dataset title | Dataset URL | Database and Identifier |
|---|---|---|---|---|
| Sheintuch L, Deitch D | 2023 | zivlab/cell_assemblies: v1.0.1 | https://doi.org/10.5281/zenodo.7635972 | Zenodo, 10.5281/zenodo.7635972 |
| Jercog PE, Abbott LF, Kandel ER | 2019 | Hippocampal Ca1 neurons recording from mice foraging in threedifferent environments over 10 days | http://doi.org/10.6080/K09W0CP7 | Collaborative Research in Computational Neuroscience, 10.6080/K09W0CP7 |

*Continued on next page*

*Continued*

| Author(s) | Year | Dataset title | Dataset URL | Database and Identifier |
|---|---|---|---|---|
| Karlsson M, Carr M, Frank LM | 2015 | Simultaneous extracellular recordings from hippocampal areas CA1 and CA3 (or MEC and CA1) from rats performing an alternation task in two W-shaped tracks that are geometrically identically but visually distinct | http://doi.org/10.6080/K0NK3BZJ | Collaborative Research in Computational Neuroscience, 10.6080/K0NK3BZJ |

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
