## [Editor Report · eLife assessment]

This study presents a new and **important** theoretical account of spatial representational drift in the hippocampus. The evidence supporting the claims is **convincing**, with a clear and accessible explanation of the phenomenon. Overall, this study will likely attract researchers exploring learning and representation in both biological and artificial neural networks.

---

## [Referee Report · Reviewer #1 (Public Review)]

The authors start from the premise that neural circuits exhibit "representational drift" -- i.e., slow and spontaneous changes in neural tuning despite constant network performance. While the extent to which biological systems exhibit drift is an active area of study and debate (as the authors acknowledge), there is enough interest in this topic to justify the development of theoretical models of drift.

The contribution of this paper is to claim that drift can reflect a mixture of "directed random motion" as well as "steady state null drift." Thus far, most work within the computational neuroscience literature has focused on the latter. That is, drift is often viewed to be a harmless byproduct of continual learning under noise. In this view, drift does not affect the performance of the circuit nor does it change the nature of the network's solution or representation of the environment. The authors aim to challenge the latter viewpoint by showing that the statistics of neural representations can change (e.g. increase in sparsity) during early stages of drift. Further, they interpret this directed form of drift as "implicit regularization" on the network.

The evidence presented in favor of these claims is concise, but on balance I find their evidence persuasive, at least in artificial network models. This paper includes a brief analysis of four independent experiments in Figure 3, which corroborates the main claims of the paper. Future work should dig deeper into the experimental data to provide a finer grained characterization. For example, in addition to quantifying the overall number of active units, it would be interesting to track changes in the signal-to-noise ratio of each place field, the widths of the place fields, et cetera.

To establish the possibility of implicit regularization in artificial networks, the authors cite convincing work from the machine learning community (Blanc et al. 2020, Li et al., 2021). Here the authors make an important contribution by translating these findings into more biologically plausible models and showing that their core assumptions remain plausible. The authors also develop helpful intuition in Figure 5 by showing a minimal model that captures the essence of their result.

---

## [Referee Report · Reviewer #2 (Public Review)]

Summary:

In the manuscript "Representational drift as a result of implicit regularization" the authors study the phenomenon of representational drift (RD) in the context of an artificial network which is trained in a predictive coding framework. When trained on a task for spatial navigation on a linear track, they found that a stochastic gradient descent algorithm led to a fast initial convergence to spatially tuned units, but then to a second very slow, yet directed drift which sparsified the representation while increasing the spatial information. They finally show that this separation of time-scales is a robust phenomenon and occurs for a number of distinct learning rules.

This is a very clearly written and insightful paper, and I think people in the community will benefit from understanding how RD can emerge in such artificial networks. The mechanism underlying RD in these models is clearly laid out and the explanation given is convincing.

It still remains unclear how this mechanism may account for the learning of multiple environments, although this is perhaps a topic for future study. The non-stationarity of the drift in this framework would seem, at first blush, to contrast with what one sees experimentally, but the authors provide compelling evidence that there are continuous changes in network properties during learning and that stationarity may be the hallmark of overfamiliarized environments. Future experimental work may further shed light on differences in RD between novel and familiar environments.

---

## [Referee Report · Reviewer #3 (Public Review)]

Summary:

Single unit neural activity tuned to environmental or behavioral variables gradually changes over time. This phenomenon, called representational drift, occurs even when all external variables remain constant, and challenges the idea that stable neural activity supports the performance of well-learned behaviors. While a number of studies have described representational drift across multiple brain regions, our understanding of the underlying mechanism driving drift is limited. Ratzon et al. propose that implicit regularization - which occurs when machine learning networks continue to reconfigure after reaching an optimal solution - could provide insights into why and how drift occurs in neurons. To test this theory, Ratzon et al. trained a recurrent neural network (RNN) trained to perform the oft-utilized linear track behavioral paradigm and compare the changes in hidden layer units to those observed in hippocampal place cells recorded in awake, behaving animals.

Ratzon et al. clearly demonstrate that hidden layer units in their model undergo consistent changes even after the task is well-learned, mirroring representational drift observed in real hippocampal neurons. They show that the drift occurs across three separate measures: the active proportion of units (referred to as sparsification), spatial information of units, and correlation of spatial activity. They continue to address the conditions and parameters under which drift occurs in their model to assess the generalizability of their findings to non-spatial tasks. Last, they investigate the mechanism through which sparsification occurs, showing that flatness of the manifold near the solution can influence how the network reconfigures. The authors suggest that their findings indicate a three stage learning process: (1) fast initial learning followed by (2) directed motion along a manifold which transitions to (3) undirected motion along a manifold.

Overall, the authors' results support the main conclusion that implicit regularization in machine learning networks mirrors representational drift observed in hippocampal place cells. Their findings promise to open new fields of inquiry into the connection between machine learning and representational drift in other, non-spatial learning paradigms, and to generate testable predictions for neural data.

Strengths:

(1) Ratzon et al. make an insightful connection between well-known phenomena in two separate fields: implicit regularization in machine learning and representational drift in the brain. They demonstrate that changes in a recurrent neural network mirror those observed in the brain, which opens a number of interesting questions for future investigation.

(2) The authors do an admirable job of writing to a large audience and make efforts to provide examples to make machine learning ideas accessible to a neuroscience audience and vice versa. This is no small feat and aids in broadening the impact of their work.

(3) This paper promises to generate testable hypotheses to examine in real neural data, e.g., that drift rate should plateau over long timescales (now testable with the ability to track single-unit neural activity across long time scales with calcium imaging and flexible silicon probes). Additionally, it provides another set of tools for the neuroscience community at large to use when analyzing the increasingly high-dimensional data sets collected today.

Weaknesses:

The revised manuscript addresses all the weaknesses outlined in my initial review. However, there is one remaining (minor) weakness regarding how "sparseness" is used and defined.

Sparseness can mean different things to different fields. For example, for engram studies, sparseness could be measured at the population level by the proportion of active cells, whereas for a physiology study, sparseness might be measured at the neuron level by the change in peak firing rate of each cell as an animal enters that cell's place field. In this manuscript, the idea of "sparseness" is introduced indirectly in the last paragraph of the introduction as "...changes in activity statistics (sparseness)...", but it is unclear from the preceding text if the referenced "activity statistics" used to define sparseness are the "fraction of active units," or their "tuning specificity," or both. While sparseness is clearly defined in the Methods section for the RNN, there is no mention of how it is defined for neural data, and spatial information is not mentioned at all. For clarity, I suggest explicitly defining sparseness for both the RNN and real neural data early in the main text, e.g. "Here, we measure sparseness in neural data by A and B, and by the analogous metric(s) of X and Y in our RNN..." This is a small but important nuance that will enhance the ease of reading for a broad neuroscience audience.

---

## [Author Response]

The following is the authors’ response to the original reviews.

**eLife assessment**
This study presents a new and valuable theoretical account of spatial representational drift in the hippocampus. The evidence supporting the claims is convincing, with a clear and accessible explanation of the phenomenon. Overall, this study will likely attract researchers exploring learning and representation in both biological and artificial neural networks.

We would like to ask the reviewers to consider elevating the assessment due to the following arguments. As noted in the original review, the study bridges two different fields (machine learning and neuroscience), and does not only touch a single subfield (representational drift in neuroscience). In the revision, we also analysed data from four different labs, strengthening the evidence and the generality of the conclusions.

**Public Reviews:**

**Reviewer #1 (Public Review):**
The authors start from the premise that neural circuits exhibit "representational drift" -- i.e., slow and spontaneous changes in neural tuning despite constant network performance. While the extent to which biological systems exhibit drift is an active area of study and debate (as the authors acknowledge), there is enough interest in this topic to justify the development of theoretical models of drift.The contribution of this paper is to claim that drift can reflect a mixture of "directed random motion" as well as "steady state null drift." Thus far, most work within the computational neuroscience literature has focused on the latter. That is, drift is often viewed to be a harmless byproduct of continual learning under noise. In this view, drift does not affect the performance of the circuit nor does it change the nature of the network's solution or representation of the environment. The authors aim to challenge the latter viewpoint by showing that the statistics of neural representations can change (e.g. increase in sparsity) during early stages of drift. Further, they interpret this directed form of drift as "implicit regularization" on the network.The evidence presented in favor of these claims is concise. Nevertheless, on balance, I find their evidence persuasive on a theoretical level -- i.e., I am convinced that implicit regularization of noisy learning rules is a feature of most artificial network models. This paper does not seem to make strong claims about real biological systems. The authors do cite circumstantial experimental evidence in line with the expectations of their model (Khatib et al. 2022), but those experimental data are not carefully and quantitatively related to the authors' model.

We thank the reviewer for pushing us to present stronger experimental evidence. We now analysed data from four different labs. Two of those are novel analyses of existing data (Karlsson et al, Jercog et al). All datasets show the same trend - increasing sparsity and increasing information per cell. We think that the results, presented in the new figure 3, allow us to make a stronger claim on real biological systems.

To establish the possibility of implicit regularization in artificial networks, the authors cite convincing work from the machine-learning community (Blanc et al. 2020, Li et al., 2021). Here the authors make an important contribution by translating these findings into more biologically plausible models and showing that their core assumptions remain plausible. The authors also develop helpful intuition in Figure 4 by showing a minimal model that captures the essence of their result.

We are glad that these translation efforts are appreciated.

In Figure 2, the authors show a convincing example of the gradual sparsification of tuning curves during the early stages of drift in a model of 1D navigation. However, the evidence presented in Figure 3 could be improved. In particular, 3A shows a histogram displaying the fraction of active units over 1117 simulations. Although there is a spike near zero, a sizeable portion of simulations have greater than 60% active units at the end of the training, and critically the authors do not characterize the time course of the active fraction for every network, so it is difficult to evaluate their claim that "all [networks] demonstrated... [a] phase of directed random motion with the low-loss space." It would be useful to revise the manuscript to unpack these results more carefully. For example, a histogram of log(tau) computed in panel B on a subset of simulations may be more informative than the current histogram in panel A.

The previous figure 3A was indeed confusing. In particular, it lumped together many simulations without proper curation. We redid this figure (now Figure 4), and added supplementary figures (Figures S1, S2) to better explain our results. It is now clear that the simulations with a large number of active units were either due to non-convergence, slow timescale of sparsification or simulations featuring label noise in which the fraction of active units is less affected. Regarding the log(tau) calculation, while it could indeed be an informative plot, it could not be calculated in a simple manner for all simulations. This is because learning curves are not always exponential, but sometimes feature initial plateaus (see also Saxe et al 2013, Schuessler et al 2020). We added a more detailed explanation of this limitation in the methods section, and we believe the current figure exemplifies the effect in a satisfactory manner.

**Reviewer #2 (Public Review):**
Summary:In the manuscript "Representational drift as a result of implicit regularization" the authors study the phenomenon of representational drift (RD) in the context of an artificial network that is trained in a predictive coding framework. When trained on a task for spatial navigation on a linear track, they found that a stochastic gradient descent algorithm led to a fast initial convergence to spatially tuned units, but then to a second very slow, yet directed drift which sparsified the representation while increasing the spatial information. They finally show that this separation of timescales is a robust phenomenon and occurs for a number of distinct learning rules.Strengths:This is a very clearly written and insightful paper, and I think people in the community will benefit from understanding how RD can emerge in such artificial networks. The mechanism underlying RD in these models is clearly laid out and the explanation given is convincing.

We thank the reviewer for the support.

Weaknesses:It is unclear how this mechanism may account for the learning of multiple environments.

There are two facets to the topic of multiple environments. First, are the results of the current paper relevant when there are multiple environments? Second, what is the interaction between brain mechanisms of dealing with multiple environments and the results of the current paper?

We believe the answer to the first question is positive. The near-orthogonality of representations between environments implies that changes in one can happen without changes in the other. This is evident, for instance, in Khatib et al and Geva et al - in both cases, drift seems to happen independently in two environments, even though they are visited intermittently and are visually similar.

The second question is a fascinating one, and we are planning to pursue it in future work. While the exact way in which the brain achieves this near-independence is an open question, remapping is one possible window into this process.

We extended the discussion to make these points clear.

The process of RD through this mechanism also appears highly non-stationary, in contrast to what is seen in familiar environments in the hippocampus, for example.

The non-stationarity noted by the reviewer is indeed a major feature of our observations, and is indeed linked to familiarity. We divide learning into three phases (now more clearly stated in Table 1 and Figure 4C). The first, rapid phase, consists of improvement of performance - corresponding to initial familiarity with the environment. The third phase, often reported in the literature of representational drift, is indeed stationary and obtained after prolonged familiarity. Our work focuses on the second phase, which is not as immediate as the first one, and can take several days. We note in the discussion that experiments which include a long familiarization process can miss this phase (see also Table 3). Furthermore, we speculate that real life is less stationary than a lab environment, and this second phase might actually be more relevant there.

**Reviewer #3 (Public Review):**
Summary:Single-unit neural activity tuned to environmental or behavioral variables gradually changes over time. This phenomenon, called representational drift, occurs even when all external variables remain constant, and challenges the idea that stable neural activity supports the performance of well-learned behaviors. While a number of studies have described representational drift across multiple brain regions, our understanding of the underlying mechanism driving drift is limited. Ratzon et al. propose that implicit regularization - which occurs when machine learning networks continue to reconfigure after reaching an optimal solution - could provide insights into why and how drift occurs in neurons. To test this theory, Ratzon et al. trained a Feedforward Network trained to perform the oft-utilized linear track behavioral paradigm and compare the changes in hidden layer units to those observed in hippocampal place cells recorded in awake, behaving animals.Ratzon et al. clearly demonstrate that hidden layer units in their model undergo consistent changes even after the task is well-learned, mirroring representational drift observed in real hippocampal neurons. They show that the drift occurs across three separate measures: the active proportion of units (referred to as sparsification), spatial information of units, and correlation of spatial activity. They continue to address the conditions and parameters under which drift occurs in their model to assess the generalizability of their findings.However, the generalizability results are presented primarily in written form: additional figures are warranted to aid in reproducibility.

We added figures, and a Github with all the code to allow full reproducibility.

Last, they investigate the mechanism through which sparsification occurs, showing that the flatness of the manifold near the solution can influence how the network reconfigures. The authors suggest that their findings indicate a three-stage learning process: (1) fast initial learning followed by (2) directed motion along a manifold which transitions to (3) undirected motion along a manifold.Overall, the authors' results support the main conclusion that implicit regularization in machine learning networks mirrors representational drift observed in hippocampal place cells.

We thank the reviewer for this summary.

However, additional figures/analyses are needed to clearly demonstrate how different parameters used in their model qualitatively and quantitatively influence drift.

We now provide additional figures regarding parameters (Figures S1, S2).

Finally, the authors need to clearly identify how their data supports the three-stage learning model they suggest.Their findings promise to open new fields of inquiry into the connection between machine learning and representational drift and generate testable predictions for neural data.Strengths:(1) Ratzon et al. make an insightful connection between well-known phenomena in two separate fields: implicit regularization in machine learning and representational drift in the brain. They demonstrate that changes in a recurrent neural network mirror those observed in the brain, which opens a number of interesting questions for future investigation.(2) The authors do an admirable job of writing to a large audience and make efforts to provide examples to make machine learning ideas accessible to a neuroscience audience and vice versa. This is no small feat and aids in broadening the impact of their work.(3) This paper promises to generate testable hypotheses to examine in real neural data, e.g., that drift rate should plateau over long timescales (now testable with the ability to track single-unit neural activity across long time scales with calcium imaging and flexible silicon probes). Additionally, it provides another set of tools for the neuroscience community at large to use when analyzing the increasingly high-dimensional data sets collected today.

We thank the reviewer for these comments. Regarding the hypotheses, these are partially confirmed in the new analyses we provide of data from multiple labs (new Figure 3 and Table 3) - indicating that prolonged exposure to the environment leads to more stationarity.

Weaknesses:(1) Neural representational drift and directed/undirected random walks along a manifold in ML are well described. However, outside of the first section of the main text, the analysis focuses primarily on the connection between manifold exploration and sparsification without addressing the other two drift metrics: spatial information and place field correlations. It is therefore unclear if the results from Figures 3 and 4 are specific to sparseness or extend to the other two metrics. For example, are these other metrics of drift also insensitive to most of the Feedforward Network parameters as shown in Figure 3 and the related text? These concerns could be addressed with panels analogous to Figures 3a-c and 4b for the other metrics and will increase the reproducibility of this work.

We note that the results from figures 3 and 4 (original manuscript) are based on abstract tasks, while in figure 2 there is a contextual notion of spatial position. Spatial position metrics are not applicable to the abstract tasks as they are simple random mapping of inputs, and there isn’t necessarily an underlying latent variable such as position. This transition between task types is better explained in the text now. In essence the spatial information and place field correlation changes are simply signatures of the movements in parameter space. In the abstract tasks their change becomes trivial, as the spatial information becomes strongly correlated with sparsity and place fields are simply the activity vectors of units. These are guaranteed to change as long as there are changes in the activity statistics. We present here the calculation of these metrics averaged over simulations for completeness.

**Author response image 1. sa4fig1:** PV correlation between training time points averaged over 362 simulations. (B) Mean SI of units normalized to first time step, averaged over 362 simulations. Red line shows the average time point of loss convergence, the shaded area represents one standard deviation.

(2) Many caveats/exceptions to the generality of findings are mentioned only in the main text without any supporting figures, e.g., "For label noise, the dynamics were qualitatively different, the fraction of active units did not reduce, but the activity of the units did sparsify" (lines 116-117). Supporting figures are warranted to illustrate which findings are "qualitatively different" from the main model, which are not different from the main model, and which of the many parameters mentioned are important for reproducing the findings.

We now added figures (S1, S2) that show this exactly. We also added a github to allow full reproduction.

(3) Key details of the model used by the authors are not listed in the methods. While they are mentioned in reference 30 (Recanatesi et al., 2021), they need to be explicitly defined in the methods section to ensure future reproducibility.

The details of the simulation are detailed in the methods sections. We also added a github to allow full reproducibility.

(4) How different states of drift correspond to the three learning stages outlined by the authors is unclear. Specifically, it is not clear where the second stage ends, and the third stage begins, either in real neural data or in the figures. This is compounded by the fact that the third stage - of undirected, random manifold exploration - is only discussed in relation to the introductory Figure 1 and is never connected to the neural network data or actual brain data presented by the authors. Are both stages meant to represent drift? Or is only the second stage meant to mirror drift, while undirected random motion along a manifold is a prediction that could be tested in real neural data? Identifying where each stage occurs in Figures 2C and E, for example, would clearly illustrate which attributes of drift in hidden layer neurons and real hippocampal neurons correspond to each stage.

Thanks for this comment, which urged us to better explain these concepts.

The different processes (reduction in loss, reduction in Hessian) happen in parallel with different timescales. Thus, there are no sharp transitions between the phases. This is now explained in the text in relation to figure 4C, where the approximate boundaries are depicted.

The term drift is often used to denote a change in representation without a change in behavior. In this sense, both the second and third phases correspond to drift. Only the third stage is stationary. This is now emphasized in the text and in the new Table 1. Regarding experimental data, apart from the new figure 3 with four datasets, we also summarize in Table 3 the relation between duration of familiarity and stationarity of the data.

**Recommendations for the authors:**
The reviewers have raised several concerns. They concur that the authors should address the specific points below to enhance the manuscript.(1) The three different phases of learning should be clearly delineated, along with how they are determined. It remains unclear in which exact phase the drift is observed.

This is now clearly explained in the new Table 1 and Figure 4C. Note that the different processes (reduction in loss, reduction in Hessian) happen in parallel with different timescales. Thus, there are no sharp transitions between the phases. This is now explained in the text in relation to figure 4C, where the approximate boundaries are depicted.

The term drift is often used to denote a change in representation without a change in behavior. In this sense, both the second and third phases correspond to drift. Only the third stage is stationary. This is now emphasized in the text and in the new Table 1. Regarding experimental data, apart from the new figure 3 with four datasets, we also summarize in Table 3 the relation between duration of familiarity and stationarity of the data.

(2) The term "sparsification" of unit activity is not fully clear. Its meaning should be more explicitly explained, especially since, in the simulations, a significant number of units appear to remain active (Fig. 3A).

We now define precisely the two measures we use - Active Fraction, and Fraction Active Units. There is a new section with an accompanying figure in the Methods section. As Figure S2 shows, the noise statistics (label noise vs. update noise) differentially affects these two measures.

(3) While the study primarily focuses on one aspect of representational drift-the proportion of active units-it should also explore other features traditionally associated with representational drift, such as spatial information and the correlation between place fields.

This absence of features is related to the abstract nature of some of the tasks simulated in our paper. In our original submission the transition between a predictive coding task to more abstract tasks was not clearly explained, creating some confusion regarding the measured metrics. We now clarified the motivation for this transition.

Both the initial simulation and the new experimental data analysis include spatial information (Figures 2,3). The following simulations (Figure 4) with many parameter choices use more abstract tasks, for which the notion of correlation between place cells and spatial information loses its meaning as there is no spatial ordering of the inputs, and every input is encountered only once. Spatial information becomes strongly correlated with the inverse of the active fraction metric. The correlation between place cells is also directly linked to increase in sparseness for these tasks.

(4) There should be a clearer illustration of how labeling noise influences learning dynamics and sparsification.

This was indeed confusing in the original submission. We removed the simulations with label noise from Figure 4, and added a supplementary figure (S2) illustrating the different effects of label noise.

(5) The representational drift observed in this study's simulations appears to be nonstationary, which differs from in vivo reports. The reasons for this discrepancy should be clarified.

We added experimental results from three additional labs demonstrating a change in activity statistics (i.e. increase in spatial information and increase in sparseness) over a long period of time. We suggest that such a change long after the environment is already familiar is an indication for the second phase, and stress that this change seems to saturate at some point, and that most drift papers start collecting data after this saturation, hence this effect was missed in previous in vivo reports. Furthermore, these effects are become more abundant with the advent on new calcium imaging methods, as the older electrophysiological regording methods did not usually allow recording of large amounts of cells for long periods of time.The new Table 3 surveys several experimental papers, emphasizing the degree of familiarity with the environment.

(6) A distinctive feature of the hippocampus is its ability to learn different spatial representations for various environments. The study does not test representational drift in this context, a topic of significant interest to the community. Whether the authors choose to delve into this is up to them, but it should at least be discussed more comprehensively, as it's only briefly touched upon in the current manuscript version.

There are two facets to the topic of multiple environments. First, are the results of the current paper relevant when there are multiple environments? Second, what is the interaction between brain mechanisms of dealing with multiple environments and the results of the current paper?

We believe the answer to the first question is positive. The near-orthogonality of representations between environments implies that changes in one can happen without changes in the other. This is evident, for instance, in Khatib et al and Geva et al - in both cases, drift seems to happen independently in two environments, even though they are visited intermittently and are visually similar.

The second question is a fascinating one, and we are planning to pursue it in future work. While the exact way in which the brain achieves this near-independence is an open question, remapping is one possible window into this process.

We extended the discussion to make these points clear.

(7) The methods section should offer more details about the neural nets employed in the study. The manuscript should be explicit about the terms "hidden layer", "units", and "neurons", ensuring they are defined clearly and not used interchangeably..

We changed the usage of these terms to be more coherent and made our code publicly available. Specifically, “units” refer to artificial networks and “neurons” to biological ones.

In addition, each reviewer has raised both major and minor concerns. These are listed below and should be addressed where possible.
**Reviewer #1 (Recommendations For The Authors):**
I recommend that the authors edit the text to soften their claims. For example:In the abstract "To uncover the underlying mechanism, we..." could be changed to "To investigate, we..."

Agree. Done

On line 21, "Specifically, recent studies showed that..." could be changed to "Specifically, recent studies suggest that..."

Agree. Done

On line 100, "All cases" should probably be softened to "Most cases" or more details should be added to Figure 3 to support the claim that every simulation truly had a phase of directed random motion.

The text was changed in accordance with the reviewer’s suggestion. In addition, the figure was changed and only includes simulations in which we expected unit sparsity to arise (without label noise). We also added explanations and supplementary figures for label noise.

Unless I missed something obvious, there is no new experimental data analysis reported in the paper. Thus, line 159 of the discussion, "a phenomenon we also observed in experimental data" should be changed to "a phenomenon that recently reported in experimental data."

We thank the reviewer for drawing our attention to this. We now analyzed data from three other labs, two of which are novel analyses on existing data. All four datasets show the same trends of sparseness with increasing spatial information. The new Figure 3 and text now describe this.

On line 179 of the Discussion, "a family of network configurations that have identical performance..." could be softened to "nearly identical performance." It would be possible for networks to have minuscule differences in performance that are not detected due to stochastic batch effects or limits on machine precision.

The text was changed in accordance with the reviewer’s suggestion.

Other minor comments:Citation 44 is missing the conference venue, please check all citations are formatted properly.

Corrected.

In the discussion on line 184, the connection to remapping was confusing to me, particularly because the cited reference (Sanders et al. 2020) is more of a conceptual model than an artificial network model that could be adapted to the setting of noisy learning considered in this paper. How would an RNN model of remapping (e.g. Low et al. 2023; Remapping in a recurrent neural network model of navigation and context inference) be expected to behave during the sparsifying portion of drift?

We now clarified this section. The conceptual model of Sanders et al includes a specific prediction (Figure 7 there) which is very similar to ours - a systematic change in robustness depending on duration of training. Regarding the Low et al model, using such mechanistic models is an exciting avenue for future research.

**Reviewer #2 (Recommendations For The Authors):**
I only have two major questions.(1) Learning multiple representations: Memory systems in the brain typically must store many distinct memories. Certainly, the hippocampus, where RD is prominent, is involved in the ongoing storage of episodic memories. But even in the idealized case of just two spatial memories, for example, two distinct linear tracks, how would this learning process look? Would there be any interference between the two learning processes or would they be largely independent? Is the separation of time scales robust to the number of representations stored? I understand that to answer this question fully probably requires a research effort that goes well beyond the current study, but perhaps an example could be shown with two environments. At the very least the authors could express their thoughts on the matter.

There are two facets to the topic of multiple environments. First, are the results of the current paper relevant when there are multiple environments? Second, what is the interaction between brain mechanisms of dealing with multiple environments and the results of the current paper?

We believe the answer to the first question is positive. The near-orthogonality of representations between environments implies that changes in one can happen without changes in the other. This is evident, for instance, in Khatib et al and Geva et al - in both cases, drift seems to happen independently in two environments, even though they are visited intermittently and are visually similar.

The second question is a fascinating one, and we are planning to pursue it in future work. While the exact way in which the brain achieves this near-independence is an open question, remapping is one possible window into this process.

We extended the discussion to make these points clear.

(2) Directed drift versus stationarity: I could not help but notice that the RD illustrated in Fig.2D is not stationary in nature, i.e. the upper right and lower left panels are quite different. This appears to contrast with findings in the hippocampus, for example, Fig.3e-g in (Ziv et al, 2013). Perhaps it is obvious that a directed process will not be stationary, but the authors note that there is a third phase of steady-state null drift. Is the RD seen there stationary? Basically, I wonder if the process the authors are studying is relevant only as a novel environment becomes familiar, or if it is also applicable to RD in an already familiar environment. Please discuss the issue of stationarity in this context.

The non-stationarity noted by the reviewer is indeed a major feature of our observations, and is indeed linked to familiarity. We divide learning into three phases (now more clearly stated in Table 1 and Figure 4C). The first, rapid, phase consists of improvement of performance - corresponding to initial familiarity with the environment. The third phase, often reported in the literature of representational drift, is indeed stationary and obtained after prolonged familiarity. Our work focuses on the second phase, which is not as immediate as the first one, and can take several days. We note in the discussion that experiments which include a long familiarization process can miss this phase (see also Table 3). Furthermore, we speculate that real life is less stationary than a lab environment, and this second phase might actually be more relevant there.

**Reviewer #3 (Recommendations For The Authors):**
Most of my general recommendations are outlined in the public review. A large portion of my comments regards increasing clarity and explicitly defining many of the terms used which may require generating more figures (to better illustrate the generality of findings) or modifying existing figures (e.g., to show how/where the three stages of learning map onto the authors' data).Sparsification is not clearly defined in the main text. As I read it, sparsification is meant to refer to the activity of neurons, but this needs to be clearly defined. For example, lines 262-263 in the methods define "sparseness" by the number of active units, but lines 116-117 state: "For label noise, the dynamics were qualitatively different, the fraction of active units did not reduce, but the activity of the units did sparsify." If the fraction of active units (defined as "sparseness") did not change, what does it mean that the activity of the units "sparsified"? If the authors mean that the spatial activity patterns of hidden units became more sharply tuned, this should be clearly stated.

We now defined precisely the two measures we use - Active Fraction, and Fraction Active Units. There is a new section with an accompanying figure in the Methods section. As Figure S2 shows, the noise statistics (label noise vs. update noise) differentially affects these two measures.

Likewise, it is unclear which of the features the authors outlined - spatial information, active proportion of units, and spatial correlation - are meant to represent drift. The authors should clearly delineate which of these three metrics they mean to delineate drift in the main text rather than leave it to the reader to infer. While all three are mentioned early on in the text (Figure 2), the authors focus more on sparseness in the last half of the text, making it unclear if it is just sparseness that the authors mean to represent drift or the other metrics as well.

The main focus of our paper is on the non-stationarity of drift. Namely that features (such as these three) systematically change in a directed manner as part of the drift process. This is in The new analyses of experimental data show sparseness and spatial information.

The focus on sparseness in the second half of the paper is because we move to more abstract These are also easy to study in the more abstract tasks in the second part of the paper. In our original submission the transition between a predictive coding task to more abstract tasks was not clearly explained, creating some confusion regarding the measured metrics. We now clarified the motivation for this transition.

It is not clear if a change in the number of active units alone constitutes "drift", especially since Geva et al. (2023) recently showed that both changes in firing rate AND place field location drive drift, and that the passage of time drives changes in activity rate (or # cells active).

Our work did not deal with purely time-dependent drift, but rather focused on experience-dependence. Furthermore, Geva et al study the stationary phase of drift, where we do not expect a systematic change in the total number of cells active. They report changes in the average firing rate of active cells in this phase, as a function of time - which does not contradict our findings.

"hidden layer", "units", and "neurons" seem to be used interchangeably in the text (e.g., line 81-85). However, this is confusing in several places, in particular in lines 83-85 where "neurons" is used twice. The first usage appears to refer to the rate maps of the hidden layer units simulated by the authors, while the second "neurons" appears to refer to real data from Ziv 2013 (ref 5). The authors should make it explicit whether they are referring to hidden layer units or actual neurons to avoid reader confusion.

We changed the usage of these terms to be more coherent. Specifically, “units” refer to artificial networks and “neurons” to biological ones.

The authors should clearly illustrate which parts of their findings support their three-phase learning theory. For example, does 2E illustrate these phases, with the first tenth of training time points illustrating the early phase, time 0.1-0.4 illustrating the intermediate phase, and 0.4-1 illustrating the last phase? Additionally, they should clarify whether the second and third stages are meant to represent drift, or is it only the second stage of directed manifold exploration that is considered to represent drift? This is unclear from the main text.

The different processes (reduction in loss, reduction in Hessian) happen in parallel with different timescales. Thus, there are no sharp transitions between the phases. This is now explained in the text in relation to figure 4C, where the approximate boundaries are depicted.

The term drift is often used to denote a change in representation without a change in behavior. In this sense, both the second and third phases correspond to drift. Only the third stage is stationary. This is now emphasized in the text and in the new Table 1. Regarding experimental data, apart from the new figure 3 with four datasets, we also summarize in Table 3 the relation between duration of familiarity and stationarity of the data.

Line 45 - It appears that the acronym ML is not defined above here anywhere.

Added.

Line 71: the ReLU function should be defined in the text, e.g., sigma(x) = x if x > 0 else 0.

Added.

106-107: Figures (or supplemental figures) to demonstrate how most parameters do not influence sparsification dynamics are warranted. As written, it is unclear what "most parameters" mean - all but noise scale. What about the learning rule? Are there any interactions between parameters?

We now removed the label noise from Figure 4, and added two supplementary figures to clearly explain the effect of parameters. Figure 4 itself was also redone to clarify this issue.

2F middle: should "change" be omitted for SI?

The panel was replaced by a new one in Figure 3.

116-119: A figure showing how results differ for label noise is warranted.

This is now done in Figure S1, S2.

124: typo, The -> the

Corrected.

127-129: This conclusion statement is the first place in the text where the three stages are explicitly outlined. There does not appear to be any support or further explanation of these stages in the text above.

We now explain this earlier at the end of the Introduction section, along with the new Table 1 and marking on Figure 4C.

132-133 seems to be more of a statement and less of a prediction or conclusion - do the authors mean "the flatness of the loss landscape in the vicinity of the solution predicts the rate of sparsification?"

We thank the reviewer for this observation. The sentence was rephrased:

Old: As illustrated in Fig. 1, different solutions in the zero-loss manifold might vary in some of their properties. The specific property suggested from theory is the flatness of the loss landscape in the vicinity of the solution.

New: As illustrated in Fig. 1, solutions in the zero-loss manifold have identical loss, but might vary in some of their properties. The authors of [26] suggest that noisy learning will slowly increase the flatness of the loss landscape in the vicinity of the solution.

135: typo, it's -> its

Corrected.

Line 135-136 "Crucially, the loss on the 136 entire manifold is exactly zero..." This appears to contradict the Figure 4A legend - the loss appears to be very high near the top and bottom edges of the manifold in 4A. Do the authors mean that the loss along the horizontal axis of the manifold is zero?

The reviewer is correct. The manifold mentioned in the sentence is indeed the horizontal axis. We changed the text and the figure to make it clearer.

Equation 6: This does not appear to agree with equation 2 - should there be an E_t term for an expectation function?

Corrected.

Line 262-263: "Sparseness means that a unit has become inactive for all inputs." This should also be stated explicitly as the definition of sparseness/sparsification in the main text.

We now define precisely the two measures we use - Active Fraction, and Fraction Active Units. There is a new section with an accompanying figure in the Methods section. As Figure S2 shows, the noise statistics (label noise vs. update noise) differentially affects these two measures.